# Pharmacology of Minor Cannabinoids at the Cannabinoid CB$_1$ Receptor: Isomer- and Ligand-Dependent Antagonism by Tetrahydrocannabivarin

**Kenneth B. Walsh [1],*** **and Andrea E. Holmes [2]**

[1]  Department of Pharmacology, Physiology & Neuroscience, School of Medicine, University of South Carolina, Columbia, SC 29209, USA

[2]  KD Pharma Group, 6934 Bioggio, Switzerland; andrea.holmes@kdpharmagroup.com

*  Correspondence: walsh@uscmed.sc.edu

**Abstract:** (1) Background: In addition to the major phytocannabinoids, *trans*-$\Delta^9$-tetrahydrocannabinol ($\Delta^9$-THC) and cannabidiol (CBD), the cannabis plant (*Cannabis sativa* L.) synthesizes over 120 additional cannabinoids that are known as minor cannabinoids. These minor cannabinoids have been proposed to act as agonists and antagonists at numerous targets including cannabinoid type 1 (CB$_1$) and type 2 (CB$_2$) receptors, transient receptor potential (TRP) channels and others. The goal of the present study was to determine the agonist effects of the minor cannabinoids: cannabinol (CBN), cannabigerol (CBG), cannabichromene (CBC), cannabitriol (CBT) and cannabidivarin (CBDV) at the CB$_1$ receptor. In addition, the CB$_1$ receptor antagonist effects of $\Delta^9$-tetrahydrocannabivarin ($\Delta^9$-THCV) were compared with its isomer $\Delta^8$-tetrahydrocannabivarin ($\Delta^8$-THCV). (2) Methods: CB$_1$ receptor activity was monitored by measuring cannabinoid activation of G protein-gated inward rectifier K$^+$ (GIRK) channels in AtT20 pituitary cells using a membrane potential-sensitive fluorescent dye assay. (3) Results: When compared to the CB$_1$ receptor full agonist WIN 55,212-2 and the partial agonist $\Delta^9$-THC, none of the minor cannabinoids caused a significant activation of G$_i$/GIRK channel signaling. However, $\Delta^9$-THCV and $\Delta^8$-THCV antagonized the effect of WIN 55,212-2 with half-maximal inhibitory concentrations (IC$_{50}$s) of 434 nM and 757 nM, respectively. $\Delta^9$-THCV antagonism of the CB$_1$ receptor was "ligand-dependent"; $\Delta^9$-THCV was more potent in inhibiting WIN 55,212-2 and 2-arachidonoylglycerol (2-AG) than $\Delta^9$-THC. (4) Conclusions: While none of the minor cannabinoids caused G$_i$/GIRK channel activation, $\Delta^9$-THCV antagonized the CB$_1$ receptor in an isomer- and ligand-dependent manner.

**Keywords:** minor cannabinoids; CB$_1$ receptor agonism and antagonism; G protein-gated inward rectifier K$^+$ channel; membrane potential; fluorescent assay

## 1. Introduction

The medicinal use of cannabis (*Cannabis sativa* L.) can be traced back thousands of years to ancient China where the plant was indicated for the treatment of rheumatic pain, constipation, gout and gynecological disorders [1,2]. Today, marijuana is approved in many countries for the relief of nausea associated with chemotherapy and for anorexia in patients with AIDS [3]. In addition, user surveys and observational studies indicate that pain management is the major reason for the medical use of cannabis [4]. Recent clinical studies also suggest that cannabis has opioid drug–sparing actions. The concomitant use of cannabis and opioids has been shown to bring about synergistic analgesic effects; thus, allowing the use of reduced opioid drug doses [5].

A major breakthrough in cannabis pharmacology came with the isolation of the phytocannabinoids cannabidiol (CBD) [6,7] and *trans*-$\Delta^9$-tetrahydrocannabinol ($\Delta^9$-THC) [8]. $\Delta^9$-THC is the main psychotropic compound produced by the plant and binds to the cannabinoid type 1 (CB$_1$) and type 2 (CB$_2$) receptors. The binding of $\Delta^9$-THC to neuronal

$CB_1$ receptors causes a stimulation of $G_i$ signaling that inhibits cAMP synthesis [9] and N-type $Ca^{2+}$ channel opening [10,11]. In addition, the $CB_1$ receptor/$G_i$ stimulation activates G protein-gated inward rectifier $K^+$ (GIRK) channels resulting in a more negative resting membrane potential and an inhibition of neurotransmitter release [10,11]. In contrast to $\Delta^9$-THC, CBD lacks psychotropic activity and has been reported to function as a weak $CB_1$ receptor antagonist [12,13] and an activator of several transient receptor potential (TRP) channels, including the TRPV1, TRPV2 and TRPA1 channels [14].

In addition to the major cannabinoids $\Delta^9$-THC and CBD, the cannabis plant produces over 120 other cannabinoids that are referred to as minor cannabinoids [15,16]. These cannabinoids are divided into neutral (e.g., cannabinol (CBN), cannabichromene (CBC) and cannabigerol (CBG)), acidic (e.g., tetrahydrocannabinolic acid (THCA) and cannabigerolic acid (CBGA)) and varinic (e.g., cannabidivarin (CBDV) and tetrahydro-cannabivarin (THCV)) compounds, and in general, are produced in smaller amounts than $\Delta^9$-THC and CBD [15,16]. Initial clinical trials suggest that some of the minor cannabinoids may be useful in the treatment of neurodegenerative diseases, epilepsy, neuropathic pain and skin disorders [17]. However, the pharmacology of the minor cannabinoids is not well understood. The proposed sites of action for these cannabinoids include the $CB_1$ and $CB_2$ receptors [18,19], TRP channels [14,20], serotonin 5-$HT_{1a}$ receptors [21,22], peroxisome proliferator-activated receptors (PPARs) [23,24] and de-orphanized receptors such as the GPR18 and GPR55 receptor [25,26]. In some studies, the minor cannabinoids CBN, CBC and CBG were reported to display weak $CB_1$ receptor agonist activity when tested in adenylyl cyclase inhibition and [$^{35}$S]GTP$\gamma$S binding assays [27,28]. In contrast, other minor cannabinoids were without $CB_1$ receptor activity in these assays [27,28]. In addition, THCV acts in vitro as a $CB_1$ receptor antagonist [29–31] but exerts both antagonist and indirect agonist effects in vivo [30]. Therefore, the goal of the present study was to determine the agonist effects of a group of minor cannabinoids on the $CB_1$ receptor by measuring the activation of GIRK channels in pituitary ATt20 cells. In addition, we sought to characterize the $CB_1$ receptor antagonist properties of THCV using the isomers $\Delta^9$-THCV and $\Delta^8$-THCV.

## 2. Materials and Methods

### 2.1. AtT20/SEPCB_1 and 5-HT_{1a} Cell Culture and Plating

The AtT20 pituitary cell line was obtained from ATCC (AtT-20/D16y-F2, CRL-1795) and grown in Dulbecco's Modified Eagle Medium (DMEM) with 10% fetal bovine serum and Pen-Strep. The AtT20 cells were stably transfected with lentivirus vectors containing either the human $CB_1$ receptor tagged with a super-ecliptic pHluorin (SEPCB_1) (a gift from Dr. Andrew Irving, [32]) or the serotonin 5-$HT_{1a}$ receptor (cDNA Resource Center, Bloomsburg, PA, USA). The tagged-$CB_1$ receptor displayed a $CB_1$ receptor signaling response similar to the unmodified receptor [33,34]. The cells were plated in poly-L-lysine-coated wells of black 96-well plates (Corning, NY, USA) (30,000 cells per well). The AtT20-SEPCB_1 and AtT20-5-$HT_{1a}$ cells were stored in an incubator at 37 °C (5% $O_2$/95% $CO_2$) and used on days 2–3 after plating.

### 2.2. GIRK Channel Fluorescent Assay

GIRK channel activation was monitored in the 96-well plates by fluorescently recording the cell membrane potential (MP) as previously described [33,34]. For the MP measurements, the AtT20-SEPCB_1 and AtT20-5-$HT_{1a}$ cells were first incubated for 30 min in a buffer solution consisting of: 132 mM NaCl, 5 mM KCl, 1 mM $CaCl_2$, 1 mM $MgCl_2$, 5 mM dextrose, 5 mM HEPES, pH 7.4 (with NaOH) and with an MP-sensitive fluorescent dye (MPSD) (FLIPR Membrane Potential kit BLUE; Molecular Devices, San Jose, CA, USA). The cells were next loaded with an MPSD in a buffer solution (as above) but containing 1 mM KCl and incubated for an additional 5 min. Fluorescent signals were then recorded in the 1 mM KCL buffer using a Synergy2 microplate reader (Biotek, Winooski, VT, USA) [33,34]. Cannabinoids and other test compounds were dissolved in DMSO at stock concentrations of 20 mM to 50 mM (final DMSO = 0.05 to 0.1%). All cannabinoids were diluted to working

concentrations in the 1 mM KCl buffer solution containing the MPSD. Due to solubility issues, concentrations of the cannabinoids above 20 µM were not tested. The cannabinoids or control solution (20 µL) were injected into each well (total volume = 220 µL) at time zero. Data points were collected at 5 s intervals over a 125 s sampling period at excitation and emission wavelengths of 520 and 560 nm. Cannabinoids that stimulate the $CB_1$ receptor cause a decrease in the fluorescent signal [33,34]. For the antagonist experiments, the cells were incubated for 2 min with various concentrations of either THCV or the $CB_1$ receptor inverse agonist/antagonist rimonabant prior to addition of the test compounds.

### 2.3. Data Analysis

Receptor antagonist concentration versus inhibition curves were obtained by normalizing the peak GIRK channel fluorescent signal in the presence of the antagonist by the response measured to a maximal concentration of the agonist alone (e.g., 5 µM WIN 55,212-2). Curves were fit using three-parameter, non-linear regression analysis:

$$E_{max}/(1 + ([drug]/IC_{50})^k),$$

where the $E_{max}$ is the maximal inhibition, $IC_{50}$ is the concentration of the compound producing a half-maxima inhibition and k is the slope factor. $IC_{50}$ values were obtained from three 96-well plates for each experimental protocol run with the antagonists and the mean $\pm$ SE determined. Curve fitting and statistical analysis were performed using Sigmaplot v8 (SPSS Inc., Chicago, IL, USA).

### 2.4. Drugs and Chemicals

The minor cannabinoids CBN, CBC, CBG, cannabitriol (CBT), CBDV, $\Delta^9$-THCV and $\Delta^8$-THCV were generously supplied by Precision Plant Molecules (Denver, CO, USA). $\Delta^9$-THC, 2-arachidonoylglycerol (2-AG) and rimonabant were purchased from Cayman Chemical Co. (Ann Arbor, MI, USA). $\Delta^9$-THC was procured using the Walsh Laboratory DEA license. 5-HT and *WAY*-100635 were obtained from Sigma–Aldrich Chemical Co. (St. Louis, MO, USA).

## 3. Results

### 3.1. Cannabinoid Agonism at the $CB_1$ Receptor

The AtT20 cells used in these experiments endogenously express GIRK (Kir3.1/3.2) channels and were stably transfected with either the human $CB_1$ receptor (AtT20-SEPCB$_1$ cells) or the human 5-HT$_{1a}$ receptor (AtT20-5-HT$_{1a}$) using lentivirus. The binding of agonists to the $CB_1$ or 5-HT$_{1a}$ receptor stimulates the dissociation of the $G_i\beta\gamma$ subunits of $G_i$ from the $G_i\alpha$ subunit. $G_i\beta\gamma$ then activates the GIRK channels causing cellular $K^+$ efflux and a concomitant decrease in the cell resting membrane potential (MP) [33,34]. Therefore, $CB_1$ receptor/Gi and 5-HT$_{1a}$ receptor/$G_i$ activities were determined in this study using a fluorescent MPSD assay [33,34].

In Figure 1, GIRK channel fluorescent responses were measured for the minor cannabinoids CBN, CBG, CBC, CBT, CBDV and THCV, as well as the major cannabinoid $\Delta^9$-THC and the synthetic cannabinoid WIN 55,212-2. The minor cannabinoids were tested at concentrations of 1, 5, 10 and 20 µM. This is consistent with concentrations of the compounds used in previous in vitro studies and approximates plasma concentrations obtained when the compounds are administered in vivo [27,28]. As expected, the full $CB_1$ receptor agonist WIN 55,212-2 (at 5 µM) produced a strong GIRK channel fluorescent response, while the partial $CB_1$ receptor agonist $\Delta^9$-THC (at 10 and 20 µM) caused a smaller fluorescent change (Figure 1). In contrast, when examined at concentrations up to 20 µM, none of the minor cannabinoids caused a significant change in the fluorescent signal when compared with control solution (Figure 1).

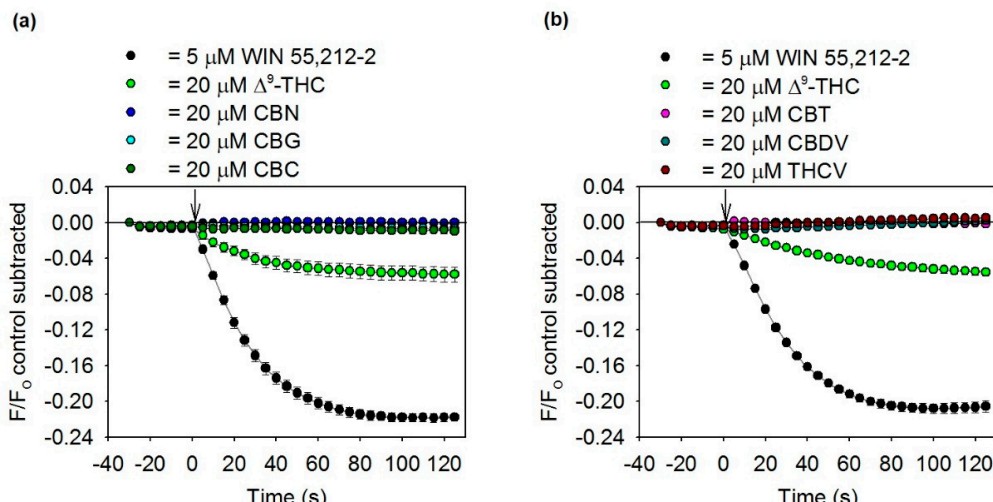

**Figure 1.** Activation of the GIRK channel fluorescent signal by cannabinoids. (**a,b**): Changes in the MPSD signal following injection of 20 μM of the indicated cannabinoids and 5 μM WIN 55,212-2 into wells containing the AtT20-SEPCB$_1$ cells. Each point represents the mean ± S.E.M. obtained in 6–8 wells. Cannabinoids were added at time zero (↓).

## 3.2. THCV Antagonism at the CB$_1$ Receptor

THCV has been reported to possess CB$_1$ receptor antagonist activity [29–31]. In the first set of experiments, the ability of THCV to inhibit the effect of WIN 55,212-2 was determined. As shown in Figure 2, $\Delta^9$-THCV inhibited the G$_i$/GIRK channel response to WIN 55,212-2 in a concentration-dependent manner with an IC$_{50}$ of 434 ± 24 nM. However, even at a concentration of 5 μM, $\Delta^9$-THCV did not produce a complete inhibition of the WIN 55,212-2-induced signal (E$_{max}$ = 79 ± 3%) (Figure 2).

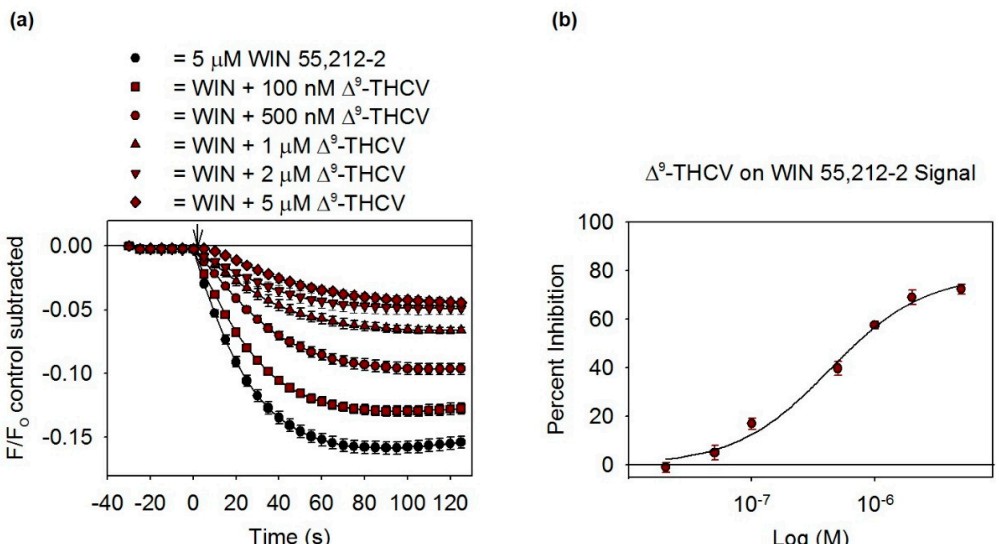

**Figure 2.** Inhibition of the GIRK channel fluorescent signal by $\Delta^9$-THCV. (**a**) Changes in the MPSD signal following injection of 5 μM WIN 55,212-2 (WIN) into wells containing the AtT20-SEPCB$_1$ cells pretreated with the indicated concentrations of $\Delta^9$-THCV. Each point represents the mean ± S.E.M. obtained in 6–8 wells. WIN was added at time zero (↓). (**b**) Concentration versus inhibition curve for $\Delta^9$-THCV on the WIN signal. The responses (as in panel a) were normalized to the GIRK channel signal measured with 5 μM WIN alone and the resulting curve fit using the 3-parameter model described in the Methods section. The IC$_{50}$ for the fitted curve was 426 nM.

$\Delta^8$-THCV is a naturally occurring isomer of $\Delta^9$-THCV that contains a double bond between carbon atoms eight and nine of the compound, rather than between carbon atoms nine and ten. $\Delta^8$-THCV also inhibited the effect of WIN 55,212-2 (IC$_{50}$ = 757 $\pm$ 35 nM) (Figure 3) but was statistically less potent than $\Delta^9$-THCV ($p < 0.002$, IC$_{50}$ of $\Delta^9$-THCV vs. $\Delta^8$-THCV).

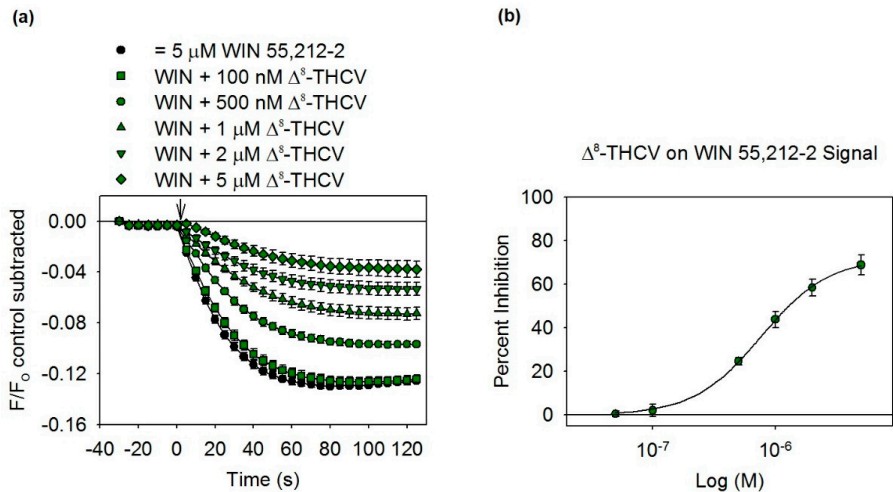

**Figure 3.** Inhibition of the GIRK channel fluorescent signal by $\Delta^8$-THCV. (**a**) Changes in the MPSD signal following injection of 5 μM WIN 55,212-2 (WIN) into wells containing the AtT20-SEPCB$_1$ cells pretreated with the indicated concentrations of $\Delta^8$-THCV. Each point represents the mean $\pm$ S.E.M. obtained in 6–8 wells. WIN was added at time zero (↓). (**b**) Concentration versus inhibition curve for $\Delta^8$-THCV on the WIN signal. The responses (as in panel a) were normalized to the GIRK channel signal measured with 5 μM WIN alone and the resulting curve fit using the 3-parameter model described in the Methods section. The IC$_{50}$ for the fitted curve was 762 nM.

The CB$_1$ receptor antagonist effects of the THCV isomers were compared with that of rimonabant (Figure 4), which is a drug developed as a CB$_1$ selective inverse agonist/antagonist.

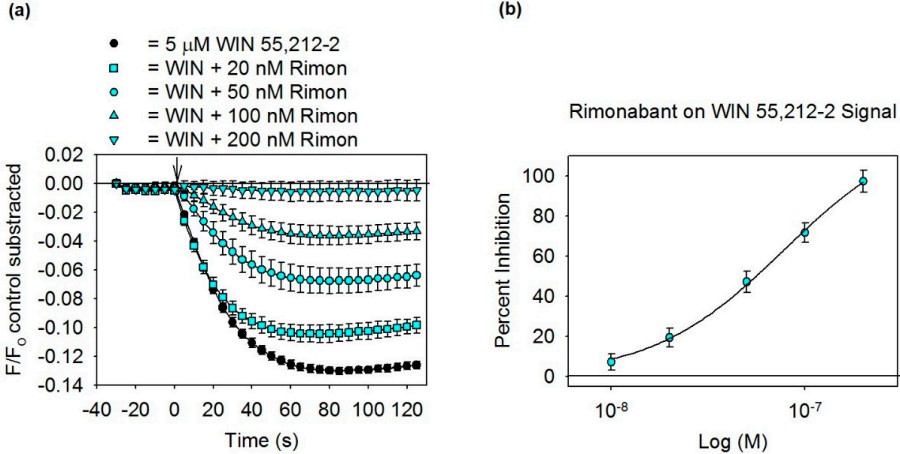

**Figure 4.** Inhibition of the GIRK channel fluorescent signal by rimonabant. (**a**) Changes in the MPSD signal following injection of 5 μM WIN 55,212-2 (WIN) into wells containing the AtT20-SEPCB$_1$ cells pretreated with the indicated concentrations of rimonabant (Rimon). Each point represents the mean $\pm$ S.E.M. obtained in 6–8 wells. WIN was added at time zero (↓). (**b**) Concentration versus inhibition curve for rimonabant on the WIN signal. The responses (as in panel **a**) were normalized to the GIRK channel signal measured with 5 μM WIN alone and the resulting curve fit using the 3-parameter model described in the Methods section. The IC$_{50}$ for the fitted curve was 76 nM.

As an antagonist in the assay, Rimonabant ($IC_{50}$ = 74 $\pm$ 5 nM) was approximately 6-fold and 10-fold more potent than $\Delta^9$-THCV and $\Delta^8$-THCV, respectively. In addition, unlike $\Delta^9$-THCV and $\Delta^8$-THCV, rimonabant produced a complete inhibition of the fluorescent signal (97 $\pm$ 6% with 200 nM rimonabant).

In Figure 5, the "ligand dependence" and receptor selectivity of $\Delta^9$-THCV antagonism was examined. To measure the "ligand dependence" of the antagonism, concentration versus inhibition curves for $\Delta^9$-THCV were obtained during $CB_1$ receptor stimulation with the endocannabinoid 2-AG and $\Delta^9$-THC. These curves were then compared with the curves obtained for $\Delta^9$-THCV inhibition of WIN 55,212-2 (as in Figure 2b). The $IC_{50}$ for $\Delta^9$-THCV inhibition of the GIRK channel fluorescent signal induced by 2-AG ($IC_{50}$ = 414 $\pm$ 19 nM) (Figure 5a) was not significantly different ($p > 0.05$) than inhibition of WIN 55,212-2 (Figure 2). In contrast, $\Delta^9$-THCV was much less potent in inhibiting $CB_1$ receptor stimulation in the presence of $\Delta^9$-THC (Figure 5b) ($IC_{50}$ = 1.20 $\pm$ 0.16 $\mu$M, $p < 0.01$ vs. $\Delta^9$-THCV inhibition of WIN 55,212-2). The THCV antagonist results are summarized in Table 1.

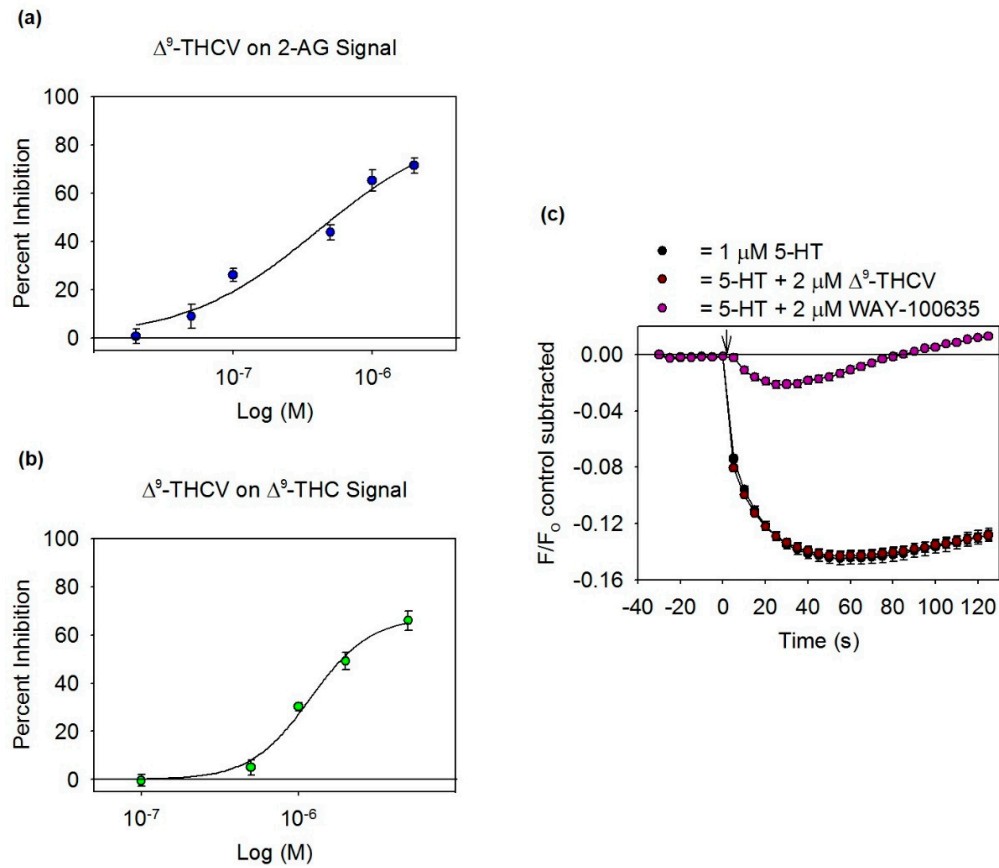

**Figure 5.** "Ligand dependence" of $\Delta^9$-THCV antagonism. (**a**,**b**) Concentration versus inhibition curves for $\Delta^9$-THCV on the 2-AG (2.5 $\mu$M) (**a**) and $\Delta^9$-THC (20 $\mu$M) (**b**) signals. The responses were normalized to the GIRK channel signal measured with 2.5 $\mu$M 2-AG or 20 $\mu$M $\Delta^9$-THC and the resulting curves fit using the 3-parameter model described in the Methods section. The $IC_{50}$s for the fitted curves were 404 nM (**a**) and 1.19 $\mu$M (**b**). (**c**) Changes in the MPSD signal following the injection of 1 $\mu$M 5-HT into wells containing the AtT20-5-HT$_{1a}$ cells pretreated with either $\Delta^9$-THCV or WAY-100635. Each point represents the mean $\pm$ S.E.M. obtained in 6–8 wells. 5-HT was added at time zero ($\downarrow$).

**Table 1.** $CB_1$ receptor antagonism by THCV.

| $CB_1$ Antagonist Assay | $IC_{50}$ [1] | Emax [1] | *p* Values for $IC_{50}$ versus $\Delta^9$-THCV on WIN |
|---|---|---|---|
| $\Delta^9$-THCV on WIN | $434 \pm 24$ nM | $79 \pm 3\%$ | |
| $\Delta^8$-THCV on WIN | $757 \pm 35$ nM | $71 \pm 4\%$ | $p < 0.002$ |
| $\Delta^9$-THCV on 2-AG | $414 \pm 19$ nM | $82 \pm 6\%$ | $p > 0.05$ |
| $\Delta^9$-THCV on $\Delta^9$-THC | $1.20 \pm 0.16$ μM | $73 \pm 4\%$ | $p < 0.01$ |

[1] Mean $\pm$ SE obtained from three plates.

In the final set of experiments, receptor selective antagonism by $\Delta^9$-THCV was determined using AtT20-5-$HT_{1a}$ cells that express the 5-$HT_{1a}$ receptor, but not the $CB_1$ receptor. As shown in Figure 5c, the application of 5-HT to the AtT20-5-$HT_{1a}$ cells resulted in a strong GIRK channel fluorescent signal which is indicative of 5-$HT_{1a}$ receptor stimulation. However, the pretreatment of the AtT20-5-$HT_{1a}$ cells with $\Delta^9$-THCV caused no change in the 5-HT-induced signal. In contrast, the 5-$HT_{1a}$ selective antagonist, WAY-100635, caused a large inhibition (>85%) in the peak 5-HT effect.

## 4. Discussion

Recent advances in the bioengineering of microbial systems to synthesize cannabinoids, along with improvements in the cannabis extraction methods, have provided a greater availability of minor cannabinoids. As such, cannabinoids including CBN, CBG, CBC and THCV are now being introduced into topical creams, beverages and edibles for consumer use. Preclinical and clinical studies suggest that some minor cannabinoids may possess neuroprotective and anti-epileptic properties [17]. For example, CBDA and CBDV display efficacy in reducing both the number and severity of seizures in animal and human studies [35–37]. Despite these findings, the molecular pharmacology of the minor cannabinoids remains largely unknown. While these compounds were first hypothesized to act via the endogenous cannabinoid system (i.e., $CB_1$ and $CB_2$ receptors, endocannabinoid synthesis and reuptake pathways, etc.), they are now known to act at numerous "off targets" including TRP channels [14,20], 5-$HT_{1a}$ receptors [21,22], PPARs [23,24], orphan receptors [25,26] and others.

In the present study, we evaluated $CB_1$ receptor agonist activity of a group of minor cannabinoids that consisted of CBN, CBG, CBC, CBT, CBDV and THCV. When compared with the full $CB_1$ receptor agonist WIN 55-212-2 and the partial agonist $\Delta^9$-THC, none of these minor cannabinoids caused any stimulation of the $CB_1$ receptor/$G_i$/GIRK channel signaling pathway (Figure 1). While previous studies have examined the effects of select minor cannabinoids on the $CB_1$ receptor, there are limited data comparing groups of cannabinoids under the same experimental conditions. Husni et al., 2014 [27], reported that $\Delta^9$-THCA, $\Delta^9$-THCV, CBG and CBDV display almost no functional activity when assayed using [$^{35}$S]GTPγS binding in HEK293 cells expressing the $CB_1$ receptor. In contrast, CBGA and CBN possess $CB_1$ receptor activity ($EC_{50}$ = 100 to 300 nM) comparable to $\Delta^9$-THC. The finding with CBN is consistent with other reports showing that this cannabinoid weakly inhibits adenylyl cyclase in $CB_1$ receptor-expressing COS cells [18,38]. Zagzoog et al., 2020 [28], compared the functional activity of a series of minor cannabinoids by measuring $CB_1$ receptor-mediated inhibition of forskolin-stimulated cAMP accumulation in CHO cells. While CBG, CBC and THCV showed partial agonist activity (compared with full agonist CP 55,940) at high nanomolar concentrations, no activity was measured with $\Delta^9$-THCA and CBDV. These results were consistent with the anti-nociceptive effects of $\Delta^9$-THC, CBG, CBC and THCV measured in the same study with C57Bl/6 mice. It should be noted that for their experiments, forskolin-stimulated cAMP accumulation was measured in the cells following 90 min of treatment with the cannabinoid ligands [28]. In contrast, we measured GIRK channel activation during a 2 min exposure to the cannabinoids. This may account for the different normalized responses of $\Delta^9$-THC in our assay (20% of the maximum WIN

55,212-2 fluorescent signal (Figure 1)) compared to their assay (56% of the maximum CP 55,940-induced inhibition) [28] and might explain why we observed no effects of CBG, CBC and THCV. Alternatively, CBG and CBC might display ligand-biased signaling toward inhibiting adenylyl cyclase versus GIRK channel activation. For example, anandamide (an endocannabinoid) is about seven times more biased toward inhibiting cAMP formation than stimulating pERK1/2 activity [39].

Several studies have established that THCV acts as a neutral antagonist at the $CB_1$ receptor. Pertwee and colleagues found that 1 μM THCV produces a rightward shift in the concentration versus the response curve for CP 55,940- and WIN 55,212-2- stimulated [$^{35}$S]GTPγS binding to mouse brain membranes [29,30]. The apparent dissociation constant ($K_B$) for this antagonism, calculated using Schild analysis, was 93 nM (CP 55,940) and 85 nM (WIN 55,212-2). In another study, $\Delta^9$-THCV at concentrations of 100 nM to 1 μM also antagonized WIN 55,212-2-stimulated [$^{35}$S]GTPγS binding to mouse brain membranes [31]. However, AM-251, a selective $CB_1$ receptor antagonist, was over 200-fold more potent as an antagonist of [$^{35}$S]GTPγS binding when compared with $\Delta^9$-THCV [31]. In addition to the antagonism of ligand-induced [$^{35}$S]GTPγS binding, THCV also prevented cannabinoid agonist inhibition of electrically-evoked contractions of the mouse isolated vas deferens [29]. Consistent with our results this antagonist effect was "ligand-dependent". THCV was more potent in antagonizing the contractile effects of WIN 55,212-2 and anandamide than the activity of $\Delta^9$-THC.

Obesity is a growing worldwide health issue that is associated with an increased risk of cardiovascular disease and type 2 diabetes. Clinical trials conducted in the United States and Europe in the early 2000s showed that the drug rimonabant (SR141716) decreases bodyweight and waist circumference in obese patients [40]. Unfortunately, rimonabant has adverse effects including anxiety, depression and an increase in suicidal thoughts [40,41]. For this reason, rimonabant was withdrawn from the market in Europe and was never approved for use in the United States. In animal studies, THCV decreases food intake and body weight, but without the adverse psychological effects of rimonabant [42]. In one clinical study, THCV decreased glucose levels and improved pancreatic insulin production in patients with type 2 diabetes [43]. Thus, further investigation of the in vitro and in vivo effects of THCV and related minor cannabinoids is certainly warranted.

**Author Contributions:** Conceptualization, K.B.W. and A.E.H.; methodology, K.B.W.; software, K.B.W.; validation, K.B.W. and A.E.H.; formal analysis, K.B.W.; investigation, K.B.W.; resources, K.B.W. and A.E.H.; data curation, K.B.W. and A.E.H.; writing—original draft preparation, K.B.W.; writing—review and editing, K.B.W. and A.E.H.; visualization, K.B.W.; supervision, K.B.W. and A.E.H.; project administration, K.B.W. and A.E.H.; funding acquisition, K.B.W. All authors have read and agreed to the published version of the manuscript.

**Funding:** This work was supported by US Public Health Service award NS-071530 and National Science Foundation award CBET-1606882 (KBW).

**Institutional Review Board Statement:** Not applicable.

**Informed Consent Statement:** Not applicable.

**Data Availability Statement:** Not applicable.

**Acknowledgments:** The authors thank Seungjin Shin and Haley Andersen for their assistance with the lentivirus transfection and the $CB_1$ receptor assay.

**Conflicts of Interest:** The authors declare no conflict of interest.

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
