# Peer review of "Pharmacology of Minor Cannabinoids at the Cannabinoid CB1 Receptor: Isomer- and Ligand-Dependent Antagonism by Tetrahydrocannabivarin"

_2813-2564, doi:10.3390/receptors1010002_

Round 1

Reviewer 1 Report

This is a very clearly written paper reporting the lack of activity of five minor phytocannabinoids at CB1 receptor. Use is made of an original, elegant and versatile method developed by the authors and described in previous publications.

Part of the results are confirmatory, for instance CBC, CBG and CBDV were previously found to display no activity at CB1 using the beta-arrestin recruitement assay (ref 28), which is also the case in the present study using another assay. However, the authors have now added CBN and CBT. As another originality, the present study compares two isomers (delta8 and delta9) of THCV for their antagonistic activity at CB1.

I suggest to publish that study after some minor improvements.

1. Page 2-3, lines 86-105: there are apparently two successive incubations with 5 mM then 1 mM KCl. Whereas I found in authors previous publications (Ref 33-34) that lowering KCl will allow more rapid extrusion of K+ ions, giving a stronger fluorescent signal, it is not clear to me how the experiments are conducted. I would be unable to reproduce the protocol. This should be clarified.

2. A table should be included with all IC50 values, which would make reading still easier. In addition, there are some differences between values given in the text and those reported in the legends to the figures (for instance 414 nM for 2-AG signal in Fig. 5a and 414 +/- 19 nM (lines 198-199). Such a difference should either disappear or be explained (variation between a typical experiment shown in figure and the mean of several experiments?).

3. As recalled in lines 129-130, activation of GIRK channel involves Gi-betagamma downstream of CB1. This is thus rather similar to beta-arrestin recruitment. As already mentioned in Point 1 above, this is coherent with data found for CBC, CBG and CBDV in this study and in ref 28. However, in the latter study, CBC and CBG displayed significant activity in the adenylate cyclase assay, indicating ligand bias for at least two phytocannabinoids. This should be discussed.

4. Rimonabant is significantly more potent  than THCV as a CB1 antagonist. As well discussed at the end of the discussion (line 272-282), THCV could be used as an anti-obesity drug lacking the secondary effects of rimonabant. Besides this difference in efficacy, the latter one is also known to be an inverse agonist of CB1. Is this also the case for THCV, or is that minor cannabinoid a neutral antagonist or an allosteric negative modulator? I do not think that the present data would allow a clear answer, but those possibilities should be discussed.

Author Response

Receptors-1827527

Reply to Reviewers

Reviewer #1:

The authors would like to express their appreciation to the reviewer for the time and effort involved in reviewing the paper and for the helpful comments supplied.  Changes/additions to the paper are indicated using “Track Changes.”

Minor concerns:

1) Page 2-3 lines 88-95. This section of the Methods has been rewritten to make the experimental protocol more understandable.  The cells were first incubated in buffer containing 5 mM KCl (+ the fluorescent dye) and then switched to buffer containing 1 mM KCl (+ the fluorescent dye).  All of the experiments were carried out in the 1 mM KCl buffer.

2) IC50 Table 1 has now been added to the paper summarizing the IC50 values from the CB1 receptor antagonism studies.  As indicated on page 3, lines 113-116, the mean ± SE values (given in Table 1 and in the text) are determined from three 96-well plates.  The IC50 in the figure legend refers to the IC50 obtained from one plate (6 to 8 wells per cannabinoid concentration).  For example, the IC50 for the data with ∆9-THCV in Figure 2 was 426 nM (from one plate).  The mean ± SE was 434 ± 24 nM (from three plates).

3) Cannabinoid signaling, page 7, lines 258-262. A statement has been added to the Discussion indicating that the minor cannabinoids CBG and CBC might display ligand bias in inhibiting cAMP formation.  A reference to biased signaling studies done with the endocannabinoid anandamide is added to the paper (reference # 39).

4) Rimonabant and THCV, lines 278-288. It is indicated on page 7, line 262, that THCV is a “neutral antagonist.”  We agree with the reviewer that the inverse agonist activity of rimonabant might account for the secondary effects of this drug.  However, behavioral studies will need to be done with other CB1 receptor neutral antagonists and inverse agonists to test this hypothesis.

Reviewer 2 Report

The aim of this study (Pharmacology of Minor Cannabinoids at the Cannabinoid CB1 Receptor: Isomer- and Ligand-Dependent Antagonism by Tetrahydrocannabivarin) was to determine the agonist effects of the minor cannabinoids at the CB1 receptor. In addition to the main cannabinoids, the cannabis plant (Cannabis sativa L.) synthesizes over 120 additional cannabinoids that are known as minor cannabinoids. Initial clinical trials suggest that some of the minor cannabinoids may be useful in the treatment of various diseases such as neurodegenerative diseases, epilepsy, neuropathic pain, and others. However, the molecular pharmacology of the minor cannabinoids is not well understood. The authors proved that the minor cannabinoid, Δ9-THCV, antagonizes the CB1 receptor and may be considered a potential obesity therapy. 

To investigate the pharmacological mechanisms of minor cannabinoids, the authors used the AtT20 pituitary cell line. The results of this study are clearly shown in the Figures and described in detail in the figure legends.

Due to the novelty of the study that minor cannabinoid Δ9-THCV can be a possible therapy for obesity or type 2 diabetes, my opinion is that this manuscript should be published in Receptors journal.

Author Response

Receptors-1827527

Reply to Reviewers

Reviewer #2:

The authors would like to express their appreciation to the reviewer for the time and effort involved in reviewing the paper and for the helpful comments supplied.  Changes/additions to the paper are indicated using “Track Changes.”

The reviewer did not list any criticisms to be addressed.